# Who Lives in the Hot Heart of the Cold Sea? A New Species of *Provanna* (Caenogastropoda: Provannidae) from the Hydrothermal Zone of Piip Volcano, Northwestern Pacific

**Ivan O. Nekhaev** 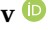

Department of Applied Ecology, St. Petersburg State University, 7/9 Universitetskaya Emb., Saint Petersburg 199034, Russia; inekhaev@gmail.com

**Abstract:** The benthic macrofauna of Arctic and subarctic seas is mainly represented by widespread species and is generally devoid of endemics. The exceptions are reduced habitats, such as cold seeps, hydrothermal areas, and wood falls, which include endemics of at least the species level. A detailed study and analysis of such endemics allows us to understand the mechanisms of colonization and to assess the age of the fauna of high-latitude marine regions. Here, the new species *Provanna annae* sp. nov. is described from the geothermal zone of the Piip volcano in the Bering Sea (subarctic Pacific) based on the morphological and molecular phylogenetic data. The new species appears to be the northernmost and one of the most shallow-water species among the known representatives of the family and is suggested to be endemic to the region. The new species is related to a group of *Provanna* species from reducing habitats off Japan. Composition of the gut content indicates that the new species bottom sediments. Bacteria are found on the gills of the mollusc and are probably symbionts that also provide nutrition. This type of mixotrophic feeding is rare in gastropods and may be a unique feature among Caenogastropoda.

**Keywords:** Bering sea; symbiosis; chemosynthesis; molecular phylogeny; bacteria

## 1. Introduction

The bottom macrofauna of Arctic and subarctic habitats is mainly represented by widespread species and is generally devoid of local endemics [1,2]. This may be due both to the difficulty of colonizing these marine regions, and to the relatively small age of the Arctic landscapes. However, the exception to this are some reduced habitats, such as cold seeps, hydrothermal areas, and wood falls, which include endemics of at least the species level [3,4]. A detailed study and analysis of such endemics allows us to obtain information about the mechanisms of colonization and the age of the fauna of high-latitude marine regions.

In the northernmost region of the Pacific Ocean, the Bering Sea, two areas with reduced habitats are known: methane seeps on the Koryak slope in the northwestern part of the sea and geothermal vents on the Piip underwater volcano in the southern part of the sea [5,6]. Species composition and patterns of occurrence of gastropods in methane seep communities are close to those in the background ecosystems, while gastropod association of Piip Volcano has features typical for some type of extreme habitats (e.g., low species diversity and high abundance) [7]. It has been suggested [7,8] that at least two of nine snail species recorded on Piip Volcano belong to taxa living only in chemosynthetic ecosystems. One of the two—*Parvaplustrum wareni* Chaban, Schepetov, Ekimova, Nekhaev et Chernyshev, 2022—was described by us earlier and was also present in the geothermal zones off Oregon, Eastern Pacific [9]. On the contrary, another species belonging to the genus *Provanna* Dall, 1918 has putative relatives off the Western Pacific coast in the Okinawa trough, on the basis of purely morphological data [7].

The goal of this study is to describe a new species of the genus *Provanna* from the geothermal zone of the Piip volcano and to discuss the biogeographic and ecological significance of this finding.

## 2. Materials and Methods

### 2.1. Study Area

Piip volcano is the northernmost (approximately 55° N and 167° E) hydrothermal region in the Pacific known to date [7,8]. The volcano is located in the central part of the Volcanologists underwater mountain massif, located on the border of the Bering Sea and the open part of the Pacific Ocean. Hydrothermal activity has been observed on two of the three summits of the volcano: the northern one at a depth range of 368–410 m and the southern one at a depth range of 464–475 m. The highest water temperature was registered in the hydrothermal manifestation of the northern summit (132.79 °C) and was significantly lower on the southern summit (10.59 °C), while the background water temperature at both summits is similar and ranges from 3.54 °C to 3.71 °C [6].

Biologically the areas of hydrothermal activity on the Piip Volcano are marked by bacterial mats and beds of vesicomyid clam *Calyptogena pacifica* Dall, 1891 (the latter have been registered only on the southern summit) [8]. The number of macro- and megabenthic species recorded in these areas are 31 species for the northern summit and 56 species for the southern summit [8]. However, the vast majority of these species are shared with background ecosystems [7,8]. Among all the benthic fauna of the Piip Volcano, only three species have been recorded exclusively in the communities of bacterial mats and clam beds: the new species of the genus *Provanna*, amphipod *Onesimoides* sp., and gastropod *Parvaplustrum wareni* [8].

### 2.2. Sampling and Sample Processing

The environment of Piip Volcano was studied during the 75th and 82nd cruises of R/V Akademik M.A. Lavrentyev organized by A.V. Zhirmunsky National Scientific Centre of Marine Biology (Vladivostok, Russia) in 2016 and 2018, respectively. The benthic fauna was sampled using a remotely operated vehicle *Commanche 18* equipped with nets (mesh diameter of 500 μm), scoop, slurp gun, manipulators, and cameras for photography and videorecording. Specimens of *Provanna* were found in seven samples (Figure 1) (Table 1). See also Nekhaev et al. [7] and Rybakova et al. [8] for more detailed information about sampling procedure. After the collection, the samples were fixed with 96% ethanol or 4% buffered formalin.

The studied material is currently stored in the collections of the Museum of the Institute of Marine Biology, A.V. Zhirmunsky National Scientific Center of Marine Biology, Far Eastern Branch, Russian Academy of Sciences, Vladivostok, Russia (MIMB) and Zoological Institute of Russian Academy of Sciences, Saint Petersburg, Russia (ZIN).

### 2.3. Morphological Studies

The external morphology and internal anatomy of snails were preliminary studied using a stereomicroscope. To obtain optical microscopy images, the snails were photographed in ethanol with Moticam X digital camera. Then, fragments of the shell, radula, and soft body of the molluscs were studied using a Zeiss Merlin Scanning Electron Microscope (SEM) equipped with a device for Energy Dispersive X-ray Spectroscopy studies. Before the SEM examination, the shells were cleaned of periostracum and dirt with an aqueous solution of sodium hypochlorite, then washed in distilled water. Radulae were extracted mechanically from the dissected specimens, then cleaned of tissue remnants using an aqueous solution of sodium hypochlorite and washed in distilled water. Prior to the SEM examination, fragments of the soft body were immersed in pure ethanol for a day, then dehydrated with hexamethyldisilizane [10]. The Energy Dispersive X-ray Spectroscopy was used for elemental analysis of the fecal pellet content. For the SEM observation, the studied samples were mounted on aluminum stubs and coated with silver.

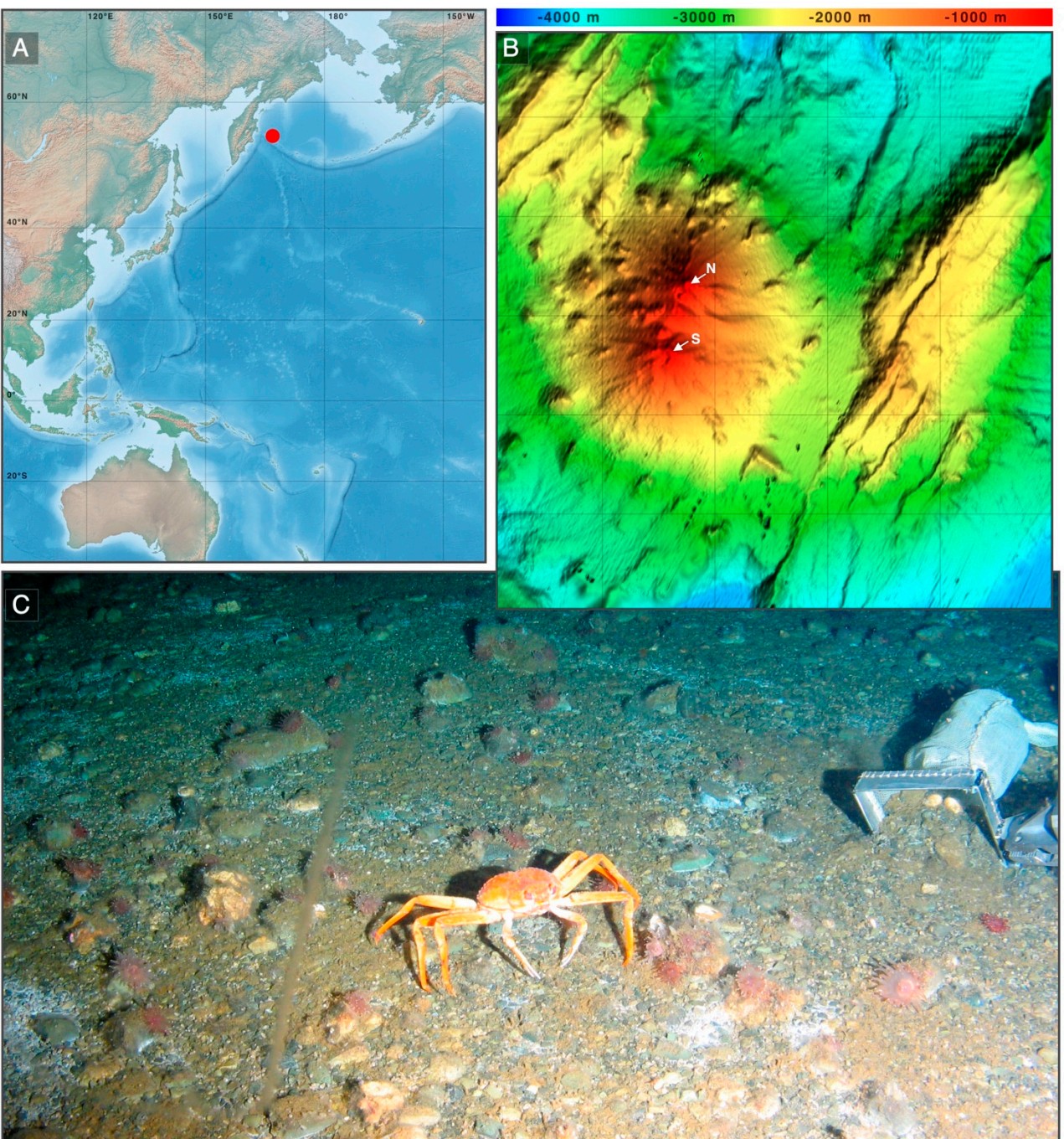

**Figure 1.** Sampling localities of new *Provanna* species. (**A**). General map of Western Pacific, red dot indicates Piip Volcano. (**B**). Bathymetrical map of Piip Volcano, arrows indicate southern (S) and northern (N) summits. (**C**). Type locality of new species on southern summit (LV82-8/3).

Shells had been measured according to the scheme by Nekhaev [11].

*2.4. Molecular Studies*

DNA was extracted from the foot of snails using DNeasy Blood & Tissue Kit (Qiagen, Venlo, The Netherlands) according to the manufacturer's protocol. Polymerase chain reaction (PCR) was performed with primer pairs Pg501L and Pg1253R for cytochrome oxidase subunit I [12], 16SAris and 16SBris for mitochondrial 16S rRNA gene fragment [13], and 28SDKF and LSU1600R for nuclear 28S rRNA gene fragment [13,14]. The total volume of PCR mixture was 20 μL, containing 1 μL of 10 μM water solution of each primer, 1 μL of

DNA solution, and 5 µL of AllTaq Master Mix (Qiagen, The Netherlands). The PCR was performed using Bio-Rad T100 thermal cycler following the protocols described in Table 2.

**Table 1.** Sampling localities of provannid snails on Piip Volcano.

| Sample | N | E | Depth, m | Date | Locality and Sampled Substrate |
|---|---|---|---|---|---|
| LV75-4/1 | 55°24.99′ | 167°16.494′ | 390 | 16 June 2016 | Northern summit, stones with bacterial mats |
| LV75-5/1 | 55°24.99′ | 167°16.494′ | 387 | 17 June 2016 | Northern summit, stones with bacterial mats |
| LV75-10/1 | 55°24.996′ | 167°16.488′ | 402 | 20 June 2016 | Northern summit, stones with bacterial mats |
| LV82-2/7 | 55°24.948′ | 167°16.596′ | 400 | 13 June 2018 | Northern summit, bacterial mat |
| LV82-3/1 | 55°22.938′ | 167°15.666′ | 469 | 14 June 2018 | Northern summit, *Calyptogena pacifica* with sediment |
| LV82-8/3 | 55°22.902′ | 167°15.672′ | 471 | 17 June 2018 | Southern summit, *Calyptogena pacifica* and stones with sediment |
| LV82-8/4 | 55°22.908′ | 167°15.684′ | 472 | 17 June 2018 | Southern summit, sediments from hydrothermal edifice |

**Table 2.** PCR conditions for amplification of gene fragments used in this study.

| Gene | Initial Denaturation | Cycle | | | | Final Annealing |
|---|---|---|---|---|---|---|
| | | Number of Cycles | Denaturation | Annealing | Elongation | |
| COI | 94 °C/120 s | 30 | 94 °C/30 s | 45 °C/30 s | 72 °C/30 s | 72 °C/40 s |
| 16S | 94 °C/300 s | 40 | 94 °C/30 s | 52 °C/30 s | 72 °C/60 s | 72 °C/600 s |
| 28S | 95 °C/180 s | 35 | 94 °C/20 s | 52 °C/60 s | 72 °C/120 s | 72 °C/600 s |

The partial gene sequences obtained by forward and reverse Sanger sequencing were reviewed and verified in Unipro UGene ver. 43.0 [15].

### 2.5. Data Analysis and Visualization

For the phylogeny reconstruction, I used reference sequences of the 16 s and COI genes obtained from Pacific species of *Provanna* and stored in Genbank. In addition to the sequences used in previous published studies [12,16,17], I used two unpublished sequences from South China Sea and one from Manus Basin (Supplementary Data S1). Unfortunately, the reference sequences for different genes have incompatible numbering, so in this study, I have reconstructed two separate phylogenies for the 16S and COI genes. The gene fragments were aligned using MAFFT algorithm with default settings. Poorly aligned positions and divergent regions were omitted in gBlocks v. 0.9 with less conservative settings. The script for this pipeline was written in the R programming language and is available at https://github.com/anisus3/Phylogeny (24 February 2023).

The final alignment was assessed by eye with Unipro Ugene. Total aligment lengths were 449 bp for 16S and 460 bp for COI. The best-fit models of nucleotide substitution evolution were selected with modelTest function of *phangorn* library for R [18]. The models were TIM2 + I for 16S and TPM2u + G + I for COI. Reconstruction of phylogenetic trees based on maximum likelihood (ML) criteria and the bootstrap analysis with 10,000 replicates were also performed in R with facilities of *phangorn* library.

Bayesian inference was calculated in multicore version of RevBayes, which a programming language based on the popular phylogenetic tool MrBayes [19]. Two parallel runs with three heated (temperature = 0.1) and one cold Markov chains each were performed for 30 million generations. Trees were sampled each 500 generations; first 15% of the trees were discarded as burn-in, and the majority-rule consensus tree was calculated from the remaining trees. The code used for Rev is available as Supplementary Material S2.

Both BI and ML trees were visualized in FigTree v. 1.4.4. Maps were made in QGis with Natural Earth map Kit. Final visuals were customized in Pixelmator Pro ver. 2.5.4 and Sketch ver. 77.

All calculations were performed on a computer with Apple M1 Pro processor working with OS X 13 in appropriate software versions.

The present work had been registered in ZooBank under the lsid (Life Science Identifier) urn:lsid:zoobank.org:pub:4F7D976D-5FD1-43F1-8B80-834C5A50C5A0.

## 3. Results

Taxonomy.

Superfamily Abyssochrysoidea Tomlin, 1927.

Family Provannidae Warén et Ponder, 1991.

Genus *Provanna* Dall, 1918.

***Provanna annae* sp. nov.**

ZooBank lsid: urn:lsid:zoobank.org:act:3089A4C2-AF53-4666-B333-C1EA19968A50.

*Provanna* sp. nov.: Nekhaev et al. [7]: 5, Figures 4A and 5C.

Type locality: Bering Sea, Northern summit of Piip Volcano, 55°22.938′ N, 167°15.666′ E, 469 m (Station LV82-8/3).

Type material: Holotype: living specimen, station LV82-8/3, MIMB 43497; Paratypes (all from type locality): 28 specimens, 2 shells, MIMB 43498; 33 specimens, MIMB 43499; 10 specimens, ZIN 63466; 12 specimens, ZIN 63467.

Other material: 2 specimens, station LV82-2/7, MIMB 43494; 2 specimens, 8 shells, station LV82-2/7, MIMB 43495; 7 specimens, station LV82-3/1, MIMB 43496; 1 specimen, station LV82-8/4, MIMB 43500; 15 specimens, station LV75-10/1, MIMB 43501; 11 specimens, station LV75-4/1, MIMB 43502; 1 specimen, station LV75-5/1, MIMB 43503.s

Etymology. The species is named after my wife Anna.

Description. Adult shell smooth, shiny, semitransparent, and of yellowish green color (Figure 2). Sculpture consists of thin straight incremental lines sometimes crossed with weak spiral Striature, visible only under SEM. Whorls convex, unevenly rounded, separated by deep suture. The base of the body whorl is weakly, but noticeably angulated in some specimens. Embryonic shell lecithotrophic and with approximately 1.2–1.3 whorls (Figure 3A,B). Nucleus with irregular granular sculpture, remaining part of protoconch covered with numerous (approximately 35) spiral ribs separated by grooves of equal size (Figure 3C,D). The upper whorls eroded in adult individuals. Aperture oval with almost 90-degree rounded angle in its upper part. Aperture height is more than one half of body whorl height. Columella curved; umbilicus covered with delicate columellar callus.

Holotype measurements: shell height = 5.0 mm, aperture height = 2.8 mm, last whorl height = 4.14 mm, shell width = 3.3 mm, and aperture width = 2.16 mm

Animals are yellowish. Proboscis large and broad; tentacles thin, thickening in their basal parts, several times longer than proboscis. Pigmented eyes absent. Males lack penis.

Radula comprises approximately 40 mature rows. Central tooth with one narrow cusp (Figure 3E,F). Lateral teeth with 3–4 broad cusps; marginal teeth spoon-like with numerous narrow denticles. Digestive tract without noticeable thickening that would correspond to the stomach. Pellets brownish, barrel-shaped, non-homogenous (Figure 4A). Morphologically, it is possible to distinguish fragments of diatom shells (Figure 4B,C,E,F) and rod-shaped structures that are considered as bacteria (Figure 4D). Energy Dispersive X-ray Spectroscopy also confirmed that in addition to carbon and oxygen, the most common element in the diet of *Provanna annae* sp. nov. was silicon (Supplementary Data S3). This indicates a high concentration of diatom parts in the studied pellets.

Ctenidium consists of thin plates. Two types of morphological structures observed in SEM images are thought to be bacteria. These are rod-shaped cells found on the surface of ctenidium lamellae, and small spheres (Figure 5A,B,D), usually located between the lamellae (Figure 5B,D).

Molecular phylogenetic analysis. Genbank accession numbers are presented in Table 3.

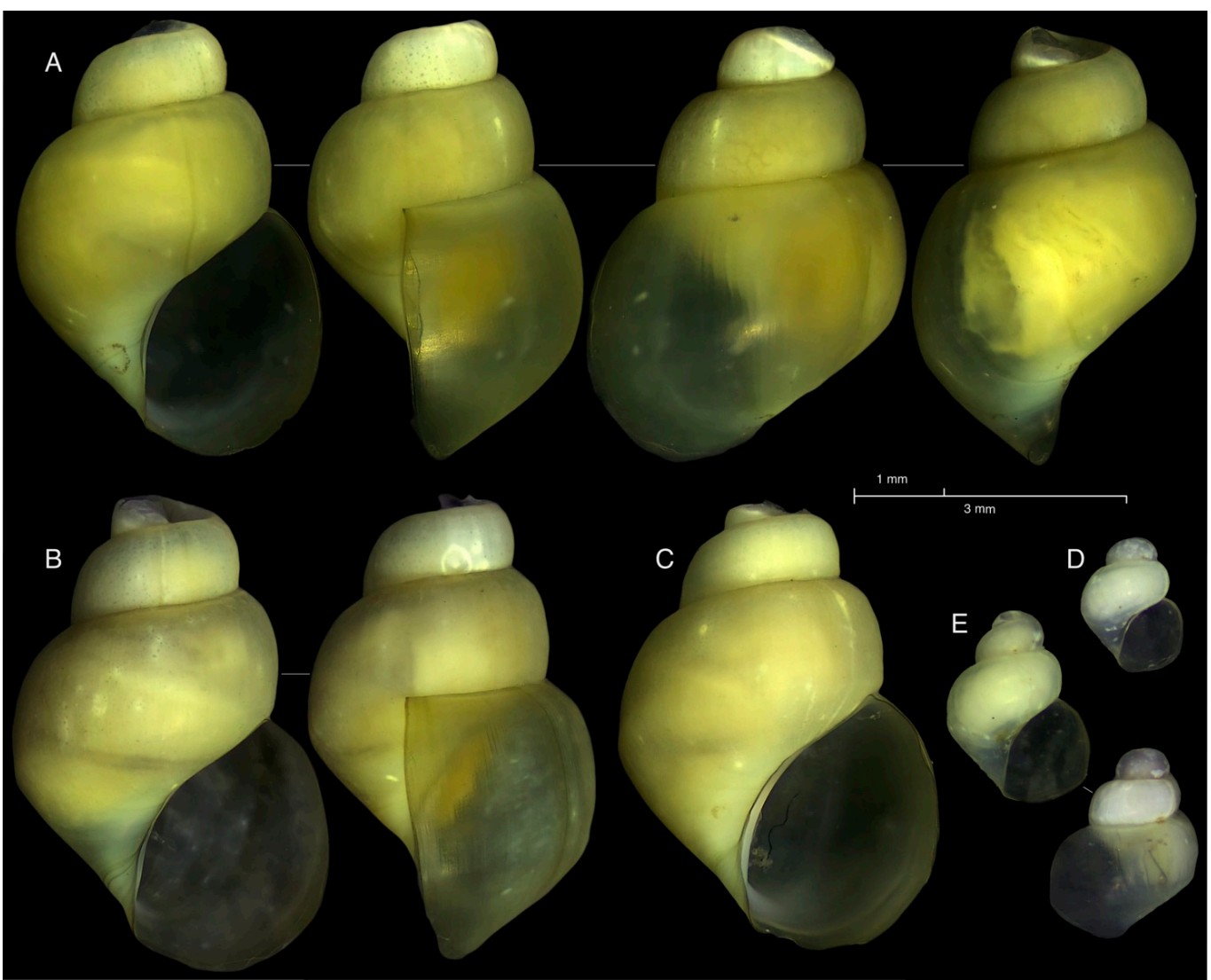

**Figure 2.** Shells of *Provanna annae* sp. nov. (A), holotype; (B,C), paratypes; (D,E), juveniles, LV75-4/1.

**Table 3.** Genbank accession numbers for molecular sequences of gastropods from Bering Sea.

| Station | Individual No. | Notes | Genbank Accession Number | | |
|---|---|---|---|---|---|
| | | | COI | 16S | 28S |
| | 3 | Paratype | - | OQ244079 | - |
| | 4 | Paratype, Figure 2B | OQ200377 | ON758771 | OQ690022 |
| LV82-8/3 | 5 | Paratype, Figure 2C | OQ200376 | ON761761 | OQ690023 |
| | 10 | Holotype, Figure 2A | - | OQ244082 | - |
| | 11 | Paratype, Figure 5A,B | - | OQ244083 | OQ690024 |
| LV75-10/1 | 1 | - | - | - | OQ690019 |

In both phylogenetic trees reconstructed using 16S and COI gene sequences, the new species is not conspecific to any of the previously known ones. No significant inconsistency was found between the Bayesian (Figure 6) and maximum likelihood (Supplementary Figure S1) analyses. However, analyses using the 16S and COI genes, respectively, place the new species as a sister group to either *Provanna glabra* Okutani, Tsuchiba et Fujikura, 1992, or to a clade comprising *Provanna kuroshimensis* Sasaki, Ogura, Watanabe et Fujikura, 2016, and *Provanna lucida* Sasaki, Ogura, Watanabe et Fujikura, 2016 (Figure 6). In any case, the

most phylogenetically close species are those living in the northwestern Pacific near Japan. In addition, the listed species and *Provanna annae* sp. nov. are characterized by the absence of sculpture or its weak development, which is not typical for most other members of the genus. It is noteworthy that the specimen from the Manus Basin, identified as *Provanna segonzaci* Warén et Ponder, 1991, was found to be conspecific with *Provanna clathrata* Sasaki, Ogura, Watanabe et Fujikura, 2016, on the basis of the 16S sequence. Considering that both species are morphologically similar, it is most likely that they are conspecific.

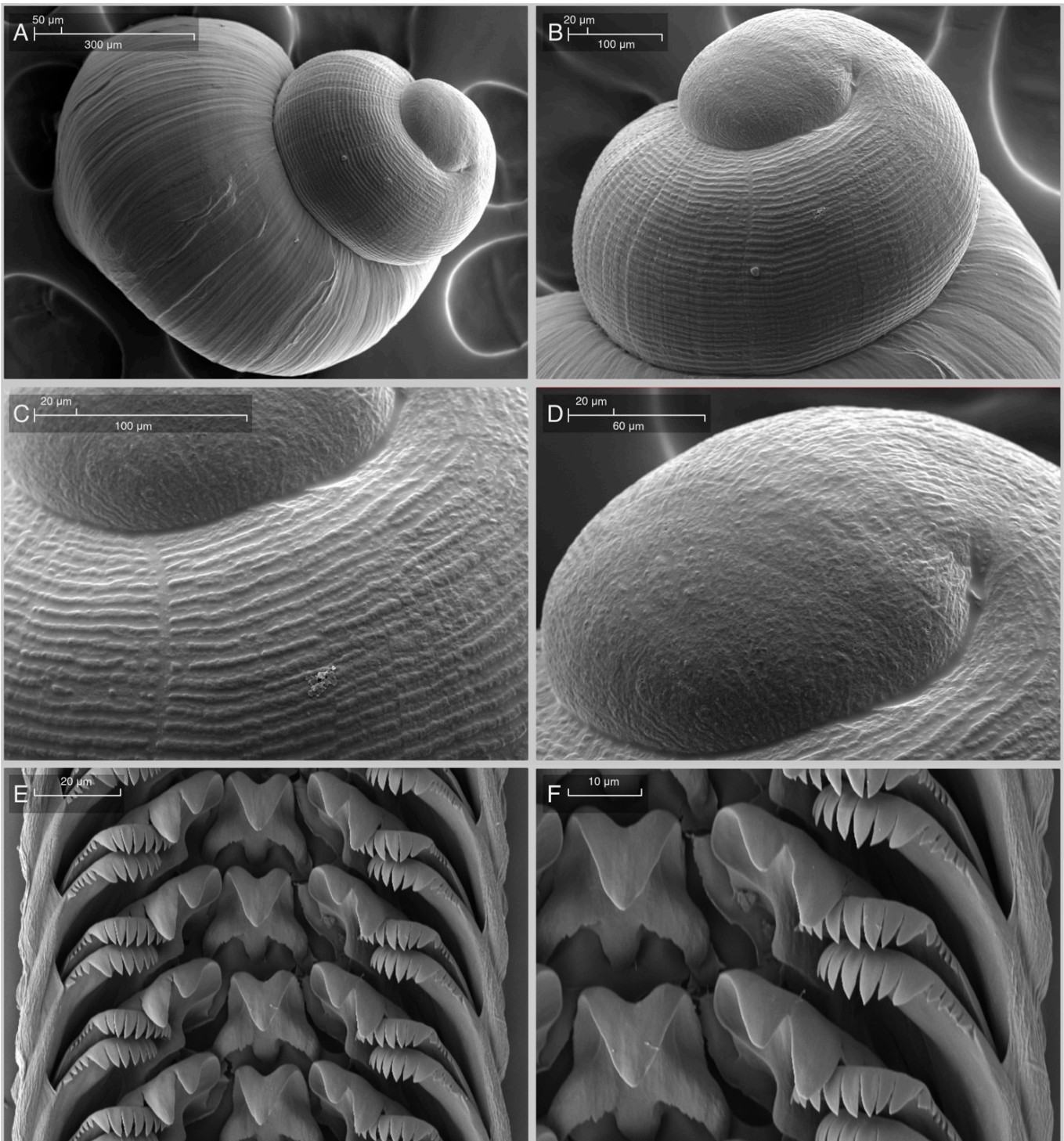

**Figure 3.** Morphological details of *Provanna annae* sp. nov. (**A**–**D**), protoconch, LV75-4/1 (same specimen as Figure 2D); (**E**,**F**), radula, paratype.

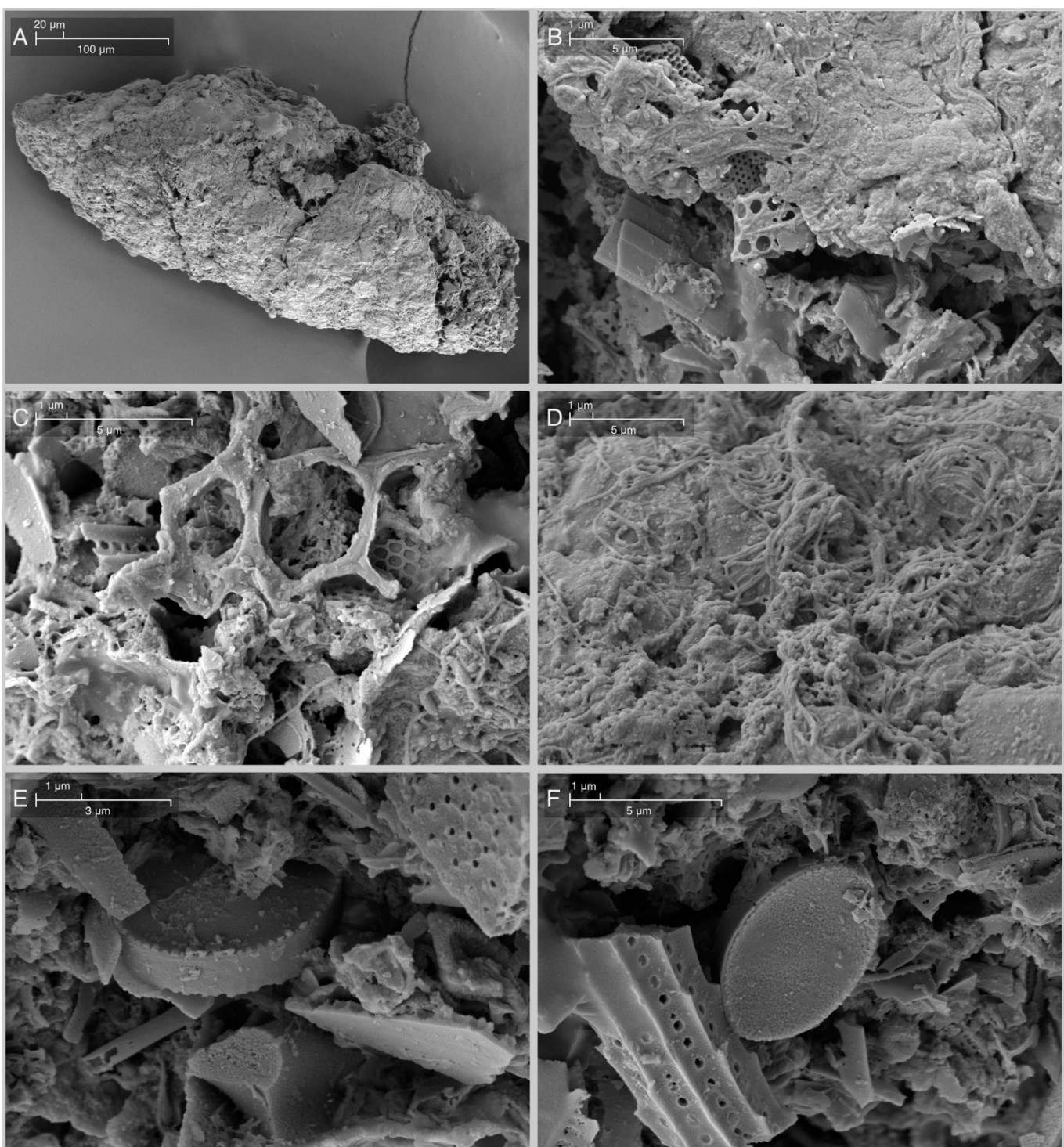

**Figure 4.** Pellets of paratype of *Provanna annae* sp. nov. (Same specimen as Figure 3E,F.). (**A**) General view of pellet; (**B**–**F**) Details of food facilities.

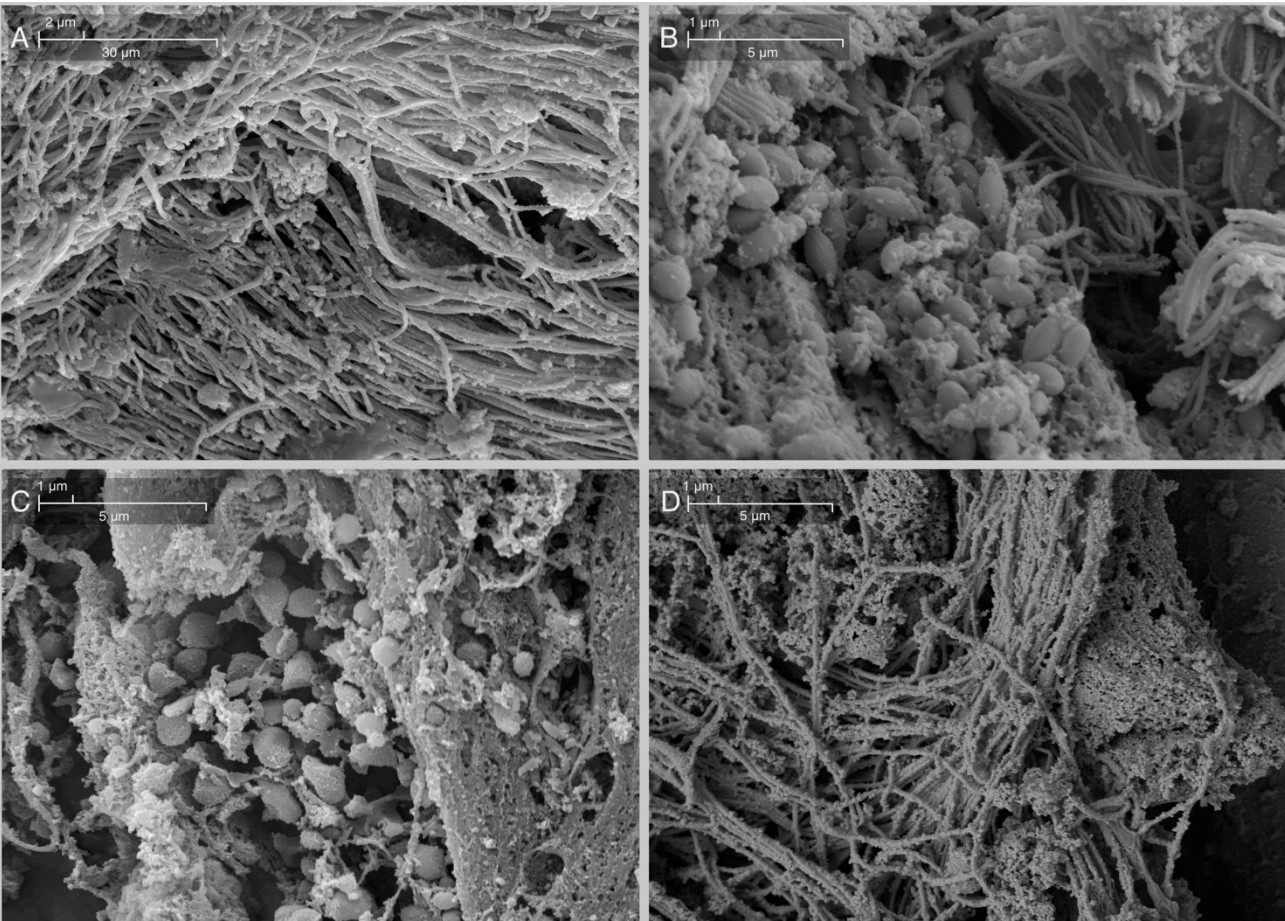

**Figure 5.** Ctenidium morphology of paratypes of *Provanna annae* sp. nov. with rod-shaped (**A**,**B**,**D**) and almost spherical bacterial cells (**B**,**C**). Figures (**A**,**B**) show one specimen, Figures (**C**,**D**) show another.

Comparison. The new species is most similar to *Provanna glabra*, *Provanna sublagbra* Sasaki, Ogura, Watanabe et Fujikura, 2016, and *Provanna kuroshimensis*; all of them can be recognized by flatter whorls, a more conical and slender shell, and by the presence of acute angle or even a sinus in the upper part of the aperture.

*Provanna lucida* Sasaki, Ogura, Watanabe et Fujikura, 2016 has smaller aperture (less than 0.5 of the body whorl height) compared to *Provanna annae* sp. nov. Additionally, some specimens of *Provanna lucida* have notable spiral cords, which has never been observed in *Provanna annae* sp. nov.

The remaining members of the genus have a sculptured shell, which distinguishes them from the new species. Additionally, *Provanna annae* differs from all members of the genus, including conchological similar ones, on the basis of the sequences of 16S and COI genes.

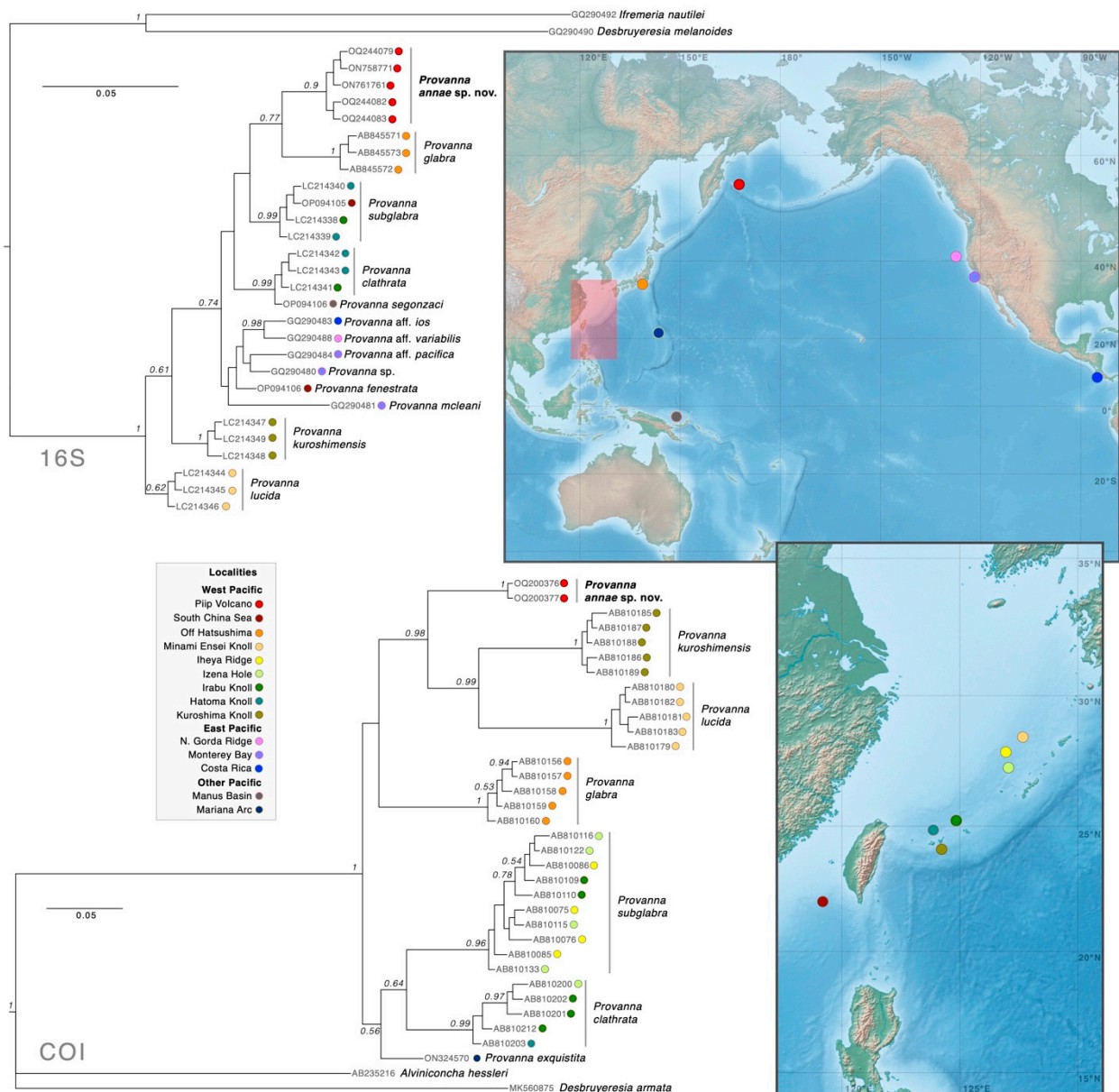

**Figure 6.** Bayesian phylogenetic trees of Pacific *Provanna* species, based on 16S and COI sequences. The maps show the geographical distribution of the specimens used in the analysis. Values of posterior probability less than 0.5 are not shown.

## 4. Discussion

### 4.1. Biogeography

The family Provannidae is widespread in the World Ocean (with the exception of the Arctic Ocean) and includes only obligate inhabitants of chemosynthetic environment [12,16,20,21]. The new species appears to be the northernmost and one of the most shallow-water species among the known representatives of the family. The group of species from hydrothermal areas off Japan was found to be the most phylogenetically close to *Provanna annae* sp. nov. (Figure 6). These areas (off Hatsushima and off Nansei-shoto area) are also the most geographically close to the Piip Volcano among the previously known records of *Provanna*. Unfortunately, nothing is known about the molluscan fauna of the hydrothermal zones near Kamchatka and the Kuril Islands, where the provannid snails can also be found.

A total of nine species of benthic macrofauna, potentially obligate of chemosynthetic communities, have been observed on the Piip Volcano. Among them, five species are

potential endemics of the far north of the Pacific Ocean: four species are known only from the Bering Sea, and one was also found in the Sea of Okhotsk [8]. Four more species were also present in the fauna of the Eastern Pacific to the south of the Bering Sea, in particular, the hydrothermal regions off the coast of Oregon and Monterey Bay. However, none of these species were shared with the fauna of the hydrothermal zones of the Western Pacific region (excluding the far north). The results of molecular phylogenetic analysis demonstrate that *Provanna annae* is strongly isolated from *Provanna* species occurring off the North American coast and has its closest relatives in Japan. However, the large phylogenetic distance between molluscs living off the Southern Asian coast and the Bering Sea likely indicates the absence of a recent faunistic exchange between the regions.

Similar phylogenetic relationships were observed in the recently described *Astyris axicostata* Kantor, Zvonareva et Krylova, 2023, which has closest relatives in the Okinawa Trough and the Sea of Japan [22]. *Astyris axicostata* is also known only from the Piip Volcano, however, this species has been observed both in geothermally active locations and in areas without marked geothermal activity [7,22]. Therefore, it is not completely clear whether this species is an obligate inhabitant of the chemosynthetic zone. Anyway, the phylogenetic relationships of *Provanna annae* and *Astyris axicostata* may indicate the presence of connections in the past between the more southern chemosynthetic regions of the East Pacific and the geothermal zone of the Piip Volcano. On the contrary, the presence of similar haplotypes in *Parvaplustrum wareni* from the Bering Sea and from Western Pacific [9] indicates the presence of recent corridors for the faunal exchange. The background fauna of the Western Bering Sea also includes species that live in the Eastern Pacific, south from the coast of California, but are limited in distribution by the Bering Sea and the Sea of Okhotsk in the western Pacific Ocean [23]. Nevertheless, a significant part of the background shelf fauna of the northwesternmost Pacific, in contrast to that of the chemosynthetic environment, is shared with more southern regions of the Western Pacific [11,24–26].

Another difference between the obligate hydrothermal fauna of Piip Volcano and the background fauna is the absence of species shared with the Arctic Ocean. The chemosynthetic fauna of the Arctic basin has been poorly studied and, in general, its research is limited only to areas of methane seeps on the shelf and regions of possible wood falls [3,4,10,27]. Virtually nothing is known about the fauna of the vast venting areas of the Gakkel ridge [28]. By analogy with the background fauna, it can be expected that the hydrothermal fauna of the Arctic basin will include taxa shared with Pacific.

Four species of obligate chemosynthetic macrofauna are found to occur both at the Piip Volcano and the more northern methane seeps of the Koryak slope of Chukotka, the species composition of which is less studied [7,8]. Nevertheless, well-known representatives of the Provannidae family are distributed within the same type of chemosynthetic habitats, usually in methane seeps or in the areas of hydrothermal venting. Therefore, it is unlikely that *Provanna annae* would also be found on the Koryak slope.

### 4.2. Feeding of Provanna

Chemosynthetic bacteria have been identified on the gills of representatives of the genera *Alviniconcha* Okutani et Ohta, 1988 and *Ifremeria* Bouchet et Warén, 1991, which are phylogenetically close to the genus *Provanna* [29]. Despite the presence of a partially reduced digestive system, there have been no reports of the presence of any food objects in it, and therefore, it is considered that these molluscs obtain nutrition only from symbiosis [30,31]. On the contrary, the developed digestive system of the *Provanna* and the presence of food particles in it led to the assumption that the molluscs are exclusively detritophagus [32,33]. The gills morphology of the new species corresponds to that of molluscs known as members of symbiotic associations with chemosynthetic bacteria [34–36].

Thus, my observations indicate that representatives of the new species are probably receiving nutrition both from bottom sediments and from the symbiosis with chemosynthetic bacteria. Perhaps this is the first detection of a mixotrophic type of nutrition in Caenogastropoda. Previously, this type of nutrition was also reported for the vetigastropod

*Lepetodrilus fucensis* McLean, 1988 from chemosynthetic zones on Juan de Fuca Ridge [35]. In such a case, symbiosis with bacteria may be optional, and the intensity of the use of bacteria as a source of nutrients depends on environmental conditions. It is very likely that the same way of nutrition can be characteristic for other *Provanna* species.

## 5. Conclusions

*Provanna annae* sp. nov. is the northernmost and possibly the shallowest known member of the genus. Its discovery allows us to suggest the presence of a closer exchange of faunas between remoted areas of hydrothermal activity in the past. The obtained results also suggest the presence of mollusks specific to the chemosynthetic environment in the regions south of the Piip Volcano, e.g., off Kamchatka and the Kuril Islands. Nevertheless, the possibility of this group of *Provanna* to penetrate the Arctic Ocean is questionable.

The discovery of bacteria on the gills of mollusks is one of the most important results of this study. This suggests a mixotrophic feeding method for *Provanna annae* sp. nov. which may be an unique feature among Caenogastropoda. However, it is necessary to consider the significant limitations of the method used. Firstly, the SEM study does not provide an opportunity to assess whether the bacteria are intracellular symbionts. Secondly, molecular identification of bacteria is desirable for more reasonable evolutionary conclusions.

**Supplementary Materials:** The following supporting information can be downloaded at: https://www.mdpi.com/article/10.3390/d15040581/s1. Supplementary Data S1: GenBank accession numbers and localities of reference sequences used for reconstruction of Provanna phylogeny. Supplementary Data S2: code for the Rev programming language used to reconstruct phylogeny by the Bayesian method. Supplementary Data S3: Results of Energy Dispersive X-ray Spectroscopy analysis of gut content. Supplementary Figure S1: Maximum likelihood phylogenetic trees based on COI and 16S gene fragments.

**Funding:** The study was supported by Russian Scientific Foundation (grant No. 21-74-00034). The SEM investigations were performed at the Interdisciplinary Center for Nanotechnology of Research Park of Saint-Petersburg State University. Molecular studies were performed at Research Park of Saint-Petersburg State University, centers "Chromas" (DNA extraction and amplification) and Centre for Molecular and Cell technologies (Sanger sequencing).

**Institutional Review Board Statement:** Not applicable.

**Data Availability Statement:** Data is contained within the article or Supplementary Materials. The R-Script for processing, aligning, and filtering DNA sequences is available at https://github.com/anisus3/Phylogeny (accessed on 24 February 2023).

**Acknowledgments:** I am grateful to the crew and members of the scientific group who participated in the 75th and 82nd voyages of the research vessel and to the staff of the Zhirmunsky National Scientific Center for Marine Biology, who performed the initial processing of the material and sent it over for further research. I also thank the staff of St. Petersburg State University Science Park for the opportunity to conduct the necessary research. Elena Rybakova (Institute of Oceanology of the Russian Academy of Sciences) provided advice on the circumstances of the collection of the snails. Ksenia Matveeva (Saint Petersburg State University) assisted in the work in the molecular laboratory. Three unpublished reference sequences of the 16S gene were provided by Shuqian Zhang (Institute of Oceanology, Chinese Academy of Sciences). I express my special gratitude to these people. I am also grateful to Varvara Bogatyreva for English editing and to three anonymous reviewers for their valuable comments to the first version of manuscript.

**Conflicts of Interest:** The author declares no conflict of interest.

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
