# Peer review of "Who Lives in the Hot Heart of the Cold Sea? A New Species of Provanna (Caenogastropoda: Provannidae) from the Hydrothermal Zone of Piip Volcano, Northwestern Pacific"

_diversity, doi:10.3390/d15040581_

Round 1
Reviewer 1 Report
In this paper the author makes the description of a new species of gastropods from deep sea.
Material and Methods
In the first section of Material and Methods the author describes the location where the new gastropod species was found. However, this description could be incorporated in the Introduction, because it does not report the sapling procedures. Moreover, the Material and Methods section is too long and could be reduced by moving at least part the description of the sampling region to the introduction.
In the "Sampling and sample processing" section the author states that samples were fixed with 96% ethanol or 4% buffered formalin (line 76). However, in the section "Morphological studies" the author says that for microscopy studies samples were preserved in ethanol. For what purpose were the formalin fixed samples used? The author mentions light microscopy photos (line 92), but there are not light microscopy images in the articles. If samples for SEM were fixed only in ethanol this is a very weak fixative for microscopy of soft tissues. This could be the reason for the lower quality of SEM images of the gill.
Results and Discussion
Line 195: I think that considering oxygen a common element in the diet is not very significant, because almost all organic molecules, and many mineral particles, contain oxygen. Therefore, only the detection of silicon is relevant to support the identification of diatom frustules in the pellets. Perhaps it would be better to say: ... in addition to carbon and oxygen, silicon was a common element in the diet ...
But, can a deep sea species captured at a depth of 390-470 m feed on live diatoms? Since these microalgae cannot live at a depth without light, the gastropods must be detritivorous and ingest frustules of dead diatoms present in sediments, which will not have much, if any, nutritional value. This problem must be properly addressed in the Discussion (line 292).
Unfortunately, SEM images do not show a global view of the ctenidium. Figure 5A does not add much to the article and could be removed. Fixation problems must have been the reason for poor preservation of soft tissues. Ethanol and formalin are not suitable fixatives for electron microscopy of soft tissues. In Figure 5C it is not clear if we are seeing the surface of the ctenidium or broken cells. The spherical and filamentous structures (Fig. 5B-D) can be bacteria, but TEM observation would be important to confirm this hypothesis. On the other hand, bacteria have also been found on the gill surface of gastropods that do not live in hydrothermal vents or other environments with reduced compounds. Therefore, the presence of bacteria on the gill does not prove that they are chemosynthetic bacteria. Can the authors provide molecular identification of the bacteria associated with the gill? This would be a very interesting addition to the article.
To conclude, I recommend publication after a proper revision of these issues.
Author Response
Q: In the first section of Material and Methods the author describes the location where the new gastropod species was found. However, this description could be incorporated in the Introduction, because it does not report the sapling procedures. Moreover, the Material and Methods section is too long and could be reduced by moving at least part the description of the sampling region to the introduction.
A: The purpose of the "Introduction" section is to pose a problem, and the formulation of the purpose of the study is not just a review of the literature. The "Materials and Methods" indicate the information necessary for the correct reproduction of the study (if necessary) and understanding of its limitations. The description of the collection area is not directly related to the formulation of the research task, it is necessary for a correct understanding of the work and ensuring the possibility of its reproduction.
Therefore, I think it is justified to put the description in the section with the Methodology. This also corresponds to the tradition.
Q: In the "Sampling and sample processing" section the author states that samples were fixed with 96% ethanol or 4% buffered formalin (line 76). However, in the section "Morphological studies" the author says that for microscopy studies samples were preserved in ethanol. For what purpose were the formalin fixed samples used? The author mentions light microscopy photos (line 92), but there are not light microscopy images in the articles. If samples for SEM were fixed only in ethanol this is a very weak fixative for microscopy of soft tissues. This could be the reason for the lower quality of SEM images of the gill.
A: This statement does not correspond to reality. Nowhere in the article was it indicated that alcohol-fixed specimens were used for morphological studies. The specimens fixed in formalin were used for morphological studies. At the same time, as a zoologist, I am obliged to correctly indicate all the material used, and not only those specimens that have been studied in more detail than the rest.
The specimens were studied using a stereomicroscope, which also uses light (not electrons) to visualize the image, this was meant by "light microscopy". To avoid further confusions, I changed the «light» to «optical».
IIndeed, both ethanol and formalin are not the best fixators for microscopic studies, so I did not try to work with transmission electron microscopy (see also comment below), which is more sensitive to the quality of fixation than scanning electron microscopy. Unfortunately, we do not yet have the opportunity to conduct a new expedition and fix the mollusks in glutaraldehyde or another acceptable fixative. Nevertheless, I believe that the scanning electron micrographs used in the article are of good quality, if the reviewer does not agree with this, then please give specific examples of artifacts that could negatively affect the results.
Q: Line 195: I think that considering oxygen a common element in the diet is not very significant, because almost all organic molecules, and many mineral particles, contain oxygen. Therefore, only the detection of silicon is relevant to support the identification of diatom frustules in the pellets. Perhaps it would be better to say: ... in addition to carbon and oxygen, silicon was a common element in the diet …
A: In my opinion, this is a reasonable remark, I have corrected it in accordance with the recommendation.
Q: But, can a deep sea species captured at a depth of 390-470 m feed on live diatoms? Since these microalgae cannot live at a depth without light, the gastropods must be detritivorous and ingest frustules of dead diatoms present in sediments, which will not have much, if any, nutritional value. This problem must be properly addressed in the Discussion (line 292).
A: Yes, of course. By themselves, diatoms are probably not a food object, but their presence is an indicator that mollusks actively consume bottom sediments, in which a large number of diatom skeletons accumulate. I clarified this in the discussion.
Q: Unfortunately, SEM images do not show a global view of the ctenidium. Figure 5A does not add much to the article and could be removed. Fixation problems must have been the reason for poor preservation of soft tissues. Ethanol and formalin are not suitable fixatives for electron microscopy of soft tissues. In Figure 5C it is not clear if we are seeing the surface of the ctenidium or broken cells. The spherical and filamentous structures (Fig. 5B-D) can be bacteria, but TEM observation would be important to confirm this hypothesis. On the other hand, bacteria have also been found on the gill surface of gastropods that do not live in hydrothermal vents or other environments with reduced compounds. Therefore, the presence of bacteria on the gill does not prove that they are chemosynthetic bacteria. Can the authors provide molecular identification of the bacteria associated with the gill? This would be a very interesting addition to the article.
A: I have already partially answered this question above. Unfortunately, we do not yet have the opportunity to fix snails in a different way. I replaced the picture 5A that caused the most dissatisfaction with another one. Nevertheless, it would be more constructive to indicate which specific artifacts, according to the reviewer, are visible in the images and how they interfere with his perception. This is more valuable than the obvious remarks about a bad fixator.
So far, we are processing molecular data on the composition of bacteria found in the gills of mollusks. According to preliminary data, there are at least several groups of chemosynthetic bacteria there. We hope that after the final processing, the results will be published after processing as part of another team of authors.
Reviewer 2 Report
I have read with interest the submitted MS of Nekhaev. It is a valuable contribution which highlights gaps in knowledge of marine biodiversity of extreme environments. In particular, the author focuses the attention on a new gastropod species that occurs in the geothermal zone of the Piip volcano, Northwestern Pacific. This manuscript is original research, and the classical morphological description and genetic analyses are appropriate approaches to support the new species investigation. The MS is well-arranged and concise, the figures are suitable, and the conclusions are appropriate. Therefore, I think that the manuscript deserves to be published. I just have some suggestions for the author:
Line 38: It would be better if the author adds to “It has been suggested” relative bibliographic references, or as was done for line 253
Lines 85 and 86: I suggest to change “red circle” with “red dot”, and “Type locality” with “Habitat type”
Line 132: I also suggest to change “was checked by eye with Unipro Ugene” with “was eye checked with Unipro Ugene”
Line 154: Please put Provanna annae always in italic.
Line 155: “Provanna sp. nov.: Nekhaev et al., 2022: 5, Figs. 4A, 5C. 155”, it is not easy to understand the meaning of this sentence, and also the meaning of the figures references. Please rewrite in a clearer and more explicit form.
Author Response
Q: Line 38: It would be better if the author adds to “It has been suggested” relative bibliographic references, or as was done for line 253
A: I add relative references
Q: Lines 85 and 86: I suggest to change “red circle” with “red dot”, and “Type locality” with “Habitat type”
Q: I changed «circle» to «dot».
A: However, I will refrain from changing the "type location" to "habitat type", since "type locality" is a very specific term
Q: Line 132: I also suggest to change “was checked by eye with Unipro Ugene” with “was eye checked with Unipro Ugene”
A: I have made language adjustments to the text
Q: Line 154: Please put Provanna annae always in italic.
A: The problem arose during the layout, apparently a straight font in this case is necessary according to the rules of the journal
Line 155: “Provanna sp. nov.: Nekhaev et al., 2022: 5, Figs. 4A, 5C. 155”, it is not easy to understand the meaning of this sentence, and also the meaning of the figures references. Please rewrite in a clearer and more explicit form
A: This is a standard reference to the mention of the species in an earlier publication
Also I add some minor additions to the text.
Reviewer 3 Report
This report of the discovery and description of a new species of Provanna provides a significant addition to the knowledge of the fauna of hydrothermal vents. I enjoyed reading the manuscript and have no major comments. I have made a number of corrections and suggestions, mostly linguistic, on the attached annotated PDF and note three more general points below.
(1) The article should be registered on Zoobank and the LSID included in the revision text.
(2) The collection of 28S rRNA sequence data is detailed but I found no mention of how these were used. If they are not used, the details about their collection (including the relevant column in Table 3) should be omitted.
(3) The references need careful checking. There are some with full journal titles and some with only abbreviated titles. Most article titles are in “title case” with capitals on most words. The journal format is for titles to be in “sentence case” (only the first letters of the sentence and of proper nouns to be in capitals). Most genus and species names are in plain font rather than italic.

Author Response
Q: (1) The article should be registered on Zoobank and the LSID included in the revision text.
A: The article will be registered there at the final submission
Q: (2) The collection of 28S rRNA sequence data is detailed but I found no mention of how these were used. If they are not used, the details about their collection (including the relevant column in Table 3) should be omitted.
A: Despite the fact that the data on this gene were not used to reconstruct the phylogeny in the article, I indicated them. This increases the volume of the article by only a couple of lines, and will not allow in the future to correctly use this data for other studies, in particular for the reconstruction of multigene phylogeny.
Q: (3) The references need careful checking. There are some with full journal titles and some with only abbreviated titles. Most article titles are in “title case” with capitals on most words. The journal format is for titles to be in “sentence case” (only the first letters of the sentence and of proper nouns to be in capitals). Most genus and species names are in plain font rather than italic.
A: I checked references and fixed some mistakes.